# Neurological Disorders Associated with WWOX Germline Mutations—A Comprehensive Overview

**DOI:** 10.3390/cells10040824

**Published:** 2021-04-07

**Authors:** Ehud Banne, Baraa Abudiab, Sara Abu-Swai, Srinivasa Rao Repudi, Daniel J. Steinberg, Diala Shatleh, Sarah Alshammery, Leszek Lisowski, Wendy Gold, Peter L. Carlen, Rami I. Aqeilan

**Affiliations:** 1The Genetic Institute, Kaplan Medical Center, Hebrew University-Hadassah Medical School, Rehovot 76100, Israel; ehudbe1@clalit.org.il; 2The Rina Mor Genetic Institute, Wolfson Medical Center, Holon 58100, Israel; 3The Concern Foundation Laboratories, The Lautenberg Center for Immunology and Cancer Research, Department of Immunology and Cancer Research-IMRIC, Hebrew University-Hadassah Medical School, Jerusalem 91120, Israel; baraa.abudiab@mail.huji.ac.il (B.A.); sara.abuswai@mail.huji.ac.il (S.A.-S.); daniel.steinberg@mail.huji.ac.il (D.J.S.); vasu369.srinivas@gmail.com (S.R.R.); diala.shatleh@mail.huji.ac.il (D.S.); 4Faculty of Medicine and Health, School of Medical Sciences and Discipline of Child and Adolescent Health, The University of Sydney, Westmead 2145, NSW, Australia; sals8341@uni.sydney.edu.au (S.A.); wendy.gold@sydney.edu.au (W.G.); 5Translational Vectorology Research Unit, Children’s Medical Research Institute, The University of Sydney, Westmead 2145, NSW, Australia; llisowski@cmri.org.au; 6Laboratory of Molecular Oncology and Innovative Therapies, Military Institute of Medicine, 04-141 Warsaw, Poland; 7Molecular Neurobiology Research Laboratory, Kids Research, Children’s Hospital at Westmead and The Children’s Medical Research Institute, Westmead 2145, NSW, Australia; 8Kids Neuroscience Centre, Kids Research, Children’s Hospital at Westmead, Westmead 2145, NSW, Australia; 9Krembil Research Institute, University Health Network and Department of Medicine, Physiology and BME, University of Toronto, Toronto, ON M5T 1M8, Canada; Peter.Carlen@uhnresearch.ca

**Keywords:** WOREE, DEE28, SCAR12, WWOX, epilepsy, personalized medicine

## Abstract

The transcriptional regulator WW domain-containing oxidoreductase (WWOX) is a key player in a number of cellular and biological processes including tumor suppression. Recent evidence has emerged associating WWOX with non-cancer disorders. Patients harboring pathogenic germline bi-allelic WWOX variants have been described with the rare devastating neurological syndromes autosomal recessive spinocerebellar ataxia 12 (SCAR12) (6 patients) and WWOX-related epileptic encephalopathy (DEE28 or WOREE syndrome) (56 patients). Individuals with these syndromes present with a highly heterogenous clinical spectrum, the most common clinical symptoms being severe epileptic encephalopathy and profound global developmental delay. Knowledge of the underlying pathophysiology of these syndromes, the range of variants of the WWOX gene and its genotype-phenotype correlations is limited, hampering therapeutic efforts. Therefore, there is a critical need to identify and consolidate all the reported variants in WWOX to distinguish between disease-causing alleles and their associated severity, and benign variants, with the aim of improving diagnosis and increasing therapeutic efforts. Here, we provide a comprehensive review of the literature on WWOX, and analyze the pathogenic variants from published and unpublished reports by collecting entries from the ClinVar, DECIPHER, VarSome, and PubMed databases to generate the largest dataset of WWOX pathogenic variants. We estimate the correlation between variant type and patient phenotype, and delineate the impact of each variant, and used GnomAD to cross reference these variants found in the general population. From these searches, we generated the largest published cohort of WWOX individuals. We conclude with a discussion on potential personalized medicine approaches to tackle the devastating disorders associated with WWOX mutations.

## 1. Introduction

Developmental and epileptic encephalopathy (DEE), previously known as early infantile epileptic encephalopathy (EIEE), is a group of severe neurological pediatric disorders affecting 3.6/100,000 live births globally [1]. DEEs are characterized by early onset, mostly intractable epilepsy, electroencephalogram (EEG) abnormalities, neurodevelopmental delay or regression, and sometimes death [2]. Onset of symptoms is during infancy, with a highly variable etiology and natural history, and with the genetic etiology being identified in half of these patients [3]. The online mendelian database (OMIM) has 93 entries for DEE, but there are probably more than 190 genes associated with some form of DEE [2]. Among these, the tumor suppressor WW domain-containing oxidoreductase (WWOX/WOX1/FOR) gene has recently been described as a DEE-causing gene [4]. Germline, bi-allelic, and pathogenic variants of WWOX have been described in the neurodevelopmental disorders WWOX-related epileptic encephalopathy (WOREE, DEE28) syndrome and spinocerebellar ataxia type 12 (SCAR12), however, despite knowing the genetic cause, the precise role of WWOX and the genotype–phenotype correlations of these disorders remain elusive. It is therefore the aim of this study to consolidate the available clinical and genetic variant data on the WWOX gene to improve our understanding of the underlying pathophysiology of the disorder, its genotype–phenotype correlation, and assist in disease management and the development of curative therapies.

## 2. Clinical Implications of WWOX Germline Variants

As introduced, variants in WWOX have been implicated in a number of disorders including SCAR12 and WOREE [4,5]. Additionally, there have been few reports of individuals with WWOX variants developing West syndrome, characterized by profound psychomotor delay and epileptic spasms with hypsarrhythmia as a specific EEG pattern [5,6]. There is also a single patient described with a disorder of sexual differentiation (DSD) [5,7].

WOREE syndrome, also known as developmental and epileptic encephalopathy 28 (DEE28), is a devastating neurological disorder characterized by refractory seizures, encephalopathy, spasticity with hyperrelexia and hypokinesia, profound developmental delay at infancy, and evident structural brain abnormalities including corpus callosum hypoplasia, progressive cerebral atrophy, white matter hyperintensity representing delayed myelination, and death by the age of 1–4 years [5]. WWOX-related West syndrome includes refractory epilepsy with the EEG pattern of hypsarrythmia, developmental delay of profound magnitude, quadriplegia, and growth restrictions including microcephaly. There is also progressive brain atrophy, suggestive of neurodegeneration [6]. Altogether, the two WWOX-related phenotypes which manifest already very early in infancy with structural brain and eye malformations are amongst the most severe and rare of all DEE forms.

SCAR12 is an autosomal recessive neurological disorder characterized by early onset seizures, delayed psychomotor development with intellectual disability, and cerebellar ataxia [4]. To date, only two families have been described with this syndrome in the literature, with both harboring a suspected pathogenic missense homozygous variant in WWOX, causing partial loss of gene function [4,8]. The affected family members presented with generalized tonic–clonic seizures before one year of age, psychomotor delay, delayed walking at 2–3 years of age, gait ataxia, and upper and lower limb ataxia. SCAR12 patients also have learning disabilities, dysarthria, gaze-evoked nystagmus without oculomotor apraxia, and diminished reflexes in the upper and lower limbs. What differentiates SCAR12 from other SCARs is the joint presence of epilepsy, spasticity, and intellectual disability. To compare this form of ataxia to other ataxias, very few dominant hereditary ataxias actually present with epilepsy. Among these are SCA17 (TBP), SCA47 (PUM1), and cerebellar atrophy with epileptic encephalopathy (FGF12). There are few similarities between these forms and SCAR12. On the other hand, the recessive forms have more similarities, including defects in several genes, including ACO2, COQ8A, KIAA0226, MTCL1, PCDH12, TDP2, and TXN2. There are also highly syndromic severe disorders that may cause the combination of ataxia and seizures, such as Zellweger syndrome [9]. Although the clinical manifestations and course of disease of WOREE syndrome are more severe than SCAR12, both disorders share many similar characteristics and can be seen as a continuum on the DEE spectrum

## 3. Molecular Characteristics

The WWOX gene is located at a common fragile site, FRA16D, on chromosome 16. The gene encodes a 46kDa WWOX protein composed of 414 amino acids consisting of two tandem WW domains at the N-terminal, WW1 and WW2, and an extended short-chain dehydrogenase/reductase (SDR) domain at the C-terminal [10,11]. The WWOX protein functions as a scaffold adaptor partnering with multiple proteins through its WW domains and modulating several protein networks [12,13] (Figure 1). Being located at a common fragile site with high frequencies of loss of heterozygosity and homozygous deletions [14], WWOX has been implicated in several types of human cancer, including breast, lung, liver, bone, and pancreatic cancers [15,16]. Several lines of evidence have associated WWOX with tumor suppression activity through its regulation of key cancer-signaling pathways, such as WNT, TGFβ1, NF-kappaB, and RTK [17,18,19,20,21,22,23,24,25], cellular metabolic pathways (HIF1α and IDH) [26,27], and response to DNA damage (ATM, p63 and p73) [28,29,30], all of which known to be directly involved in tumorigenesis. However, more recent studies associate WWOX with the nervous system, and its impaired function with numerous neurological disorders [31].

## 4. WWOX Expression in the Nervous System

The precise role of WWOX in the central and peripheral nervous system is limited. In humans, WWOX mRNA is expressed in all tissues, although it is most prominently expressed in the thyroid and some parts of the brain, for example, the cerebrum and cerebellum (https://gtexportal.org/, 1 April 2021). Murine WWOX is shown to be expressed in many regions of the nervous system, with differential WWOX protein expression being observed between the adult and embryonic stages of development [32,33]. Specifically, WWOX is predominantly detected in the peripheral nerve peduncles, brain stem, and spinal cord during the middle and late embryonic development stages of murine development. There is a decrease in expression postnatally, suggesting a role in neuronal cell differentiation. In the adult murine nervous system, WWOX is mainly expressed in the epithelial cells of the choroid plexus and in ependymal cells, with a low to moderate expression in regions such as the white matter tracts of the corpus callosum, striatum, and cerebral peduncle [33].

A detailed analysis of human WWOX mRNA expression in the CNS has been recently reviewed elsewhere [34]. In brief, WWOX mRNA expression is prominent during the early embryonic stages and decreases gradually with fetal development [34]. Upon birth, there is an increase in WWOX expression that remains stable throughout adulthood in all brain regions. In the human adult brain, WWOX is expressed in the neurons and astrocytes of the frontal and occipital cortex, the neurons of the caudate nucleus, and in the pons and the medulla [32,35], implicating WWOX involvement in CNS functions.

## 5. Modeling WWOX Deficiency in Rodents Reveals an Epileptic Phenotype

Several animal models of WWOX deficiency have been described in the literature, however, two models in particular report phenocopy epilepsy. Suzuki et al. reported that a spontaneous deletion of 13-bp in exon 9 within the WWOX gene of the lde/lde (lethal dwarfism with epilepsy) rat was associated with dwarfism, ataxic gait, high incidence of epileptic seizures, and postnatal lethality [36] providing direct evidence for the role of WWOX in CNS biology. The WWOX knockout (KO) mouse model exhibits features that phenocopy some symptoms observed in individuals with WWOX variants such as epilepsy, structural brain malformation, and learning impairments [4,37,38,39]. The full spectrum of the model includes growth retardation, severe metabolic disorder (hypoglycemia, hypolipidemia, impaired steroidogenesis, and bone metabolic phenotypes), spontaneous and audiogenic seizures, gait ataxia, severe motor incoordination, imbalance, and premature death by 2 to 3 weeks after birth [4,37,38,39]. The brains of WWOX KO mice show cerebral malformations, microcephaly, incomplete hemisphere separation, neuronal disorganization, heterotopia, defective cerebellar midline fusion, and the degeneration of many neural cell types (Figure 2). Heterotopia (the ectopic displacement of grey matter in the brain) is well known to be associated with seizures [40]. Reduced GABAergic interneurons (PV and NPY) and activation or hypertrophy of microglia and astrocytes in the hippocampus of WWOX KO mice is also consistent with an epileptic phenotype.

Transcriptional analysis of neurospheres derived from WWOX KO neural stem cells reveals alterations in the expression of genes related to neurological disorders, CNS development, and epilepsy [41]. Furthermore, transcriptomic analysis in WWOX-depleted human neural progenitor cells demonstrates that WWOX downregulation significantly alters the expression of genes involved in neuron migration and cytoskeleton organization [20,42]. Despite these studies, it is evident that detailed studies uncovering the molecular and cellular roles of WWOX in pediatric epilepsy are still lacking.

## 6. WWOX Impairment in Neurological Disorders

There is an emerging role of WWOX in a number of neurological disorders including early-onset epilepsy, autism spectrum disorder (ASD), multiple sclerosis (MS), and Alzheimer’s disease (Figure 2). However, the underlying mechanism of WWOX function is still currently unknown.

### 6.1. WWOX in Alzheimer’s Disease

Mutations in WWOX have recently been found to be a significant risk factor for disease development in Alzheimer’s disease, an adult-onset neurodegenerative disorder [43]. Furthermore, decreased WWOX protein levels have been observed in Alzheimer’s disease patients compared to age-matched healthy controls [31,44]. In the hippocampus of Alzheimer’s disease patients, an increase in the phosphorylation of the microtubule-associated protein, tau, was reported in a glycogen synthase kinase 3β (GSK-3β)-dependent pathway [44]. WWOX was found to physically interact with and inhibit GSK-3β, preventing tau hyperphosphorylation and transformation into neurofibrillary tangles (NFTs), leading to neuronal loss [45]. WWOX was also shown to promote neural differentiation by suppressing GSK-3β activity, resulting in an increase of the affinity to microtubules of tau, allowing microtubule assembly to prompt neuronal differentiation [45] (Figure 1). It has also been demonstrated that WWOX physically interacts with TPC6AΔ, an aggregated vesicle-trafficking protein isoform that has a critical role in causing caspase activation, tau aggregation, and Aβ generation in patients with Alzheimer’s disease, blocking its self-aggregation [46]. Consequent to WWOX loss, TPC6AΔ undergoes polymerization leading to the aggregation of TGF-β1-induced antiapoptotic factor (TIAF1) caspase activation, which causes amyloid precursor protein (APP) degradation, leading to the generation of amyloid β and the formation of the neurofibrillary tangles (NFTs), causing neurodegeneration (Figure 1) [47,48].

### 6.2. WWOX in Multiple Sclerosis

Multiple sclerosis (MS) is an immune-mediated disorder that affects the CNS [49]. It is a demyelinating disease that attacks the myelin sheaths covering the axons of nerve cells in the brain and the spinal cord. These attacks cause myelin sheath destruction, which can further lead to axonal damage and loss, inducing many severe neurological disabilities [50]. Lately, a study published by Jäkel et al. using single nucleus RNA-sequencing on post-mortem tissues of 4 MS individuals and 5 healthy controls determined that an alteration in oligodendrocyte population heterogeneity between MS individuals and the controls and a shift in the transcription profile of the different oligodendrocyte sub-clusters in MS tissues. Markedly, in their differential gene expression analysis, WWOX levels were shown to be reduced in the different types of MS lesions, mostly in the chronic active lesions, compared to the controls [51].

Furthermore, several lines of studies have listed WWOX as one of the susceptible genes associated with MS. A genome-wide association study (GWAS) conducted by Beecham et al. on 14,498 individuals with MS and 24,091 healthy controls and joined with previous independent GWAS data from 14,802 MS individuals and 26,703 healthy controls identified 48 susceptibility variants, one of which was the rs12149527 WWOX variant with a joint *p =* 3.3 × 10^−11^ and an odds ratio (OR) of 1.08 [52]. WWOX rs7201683, another variant, was shown to have a higher frequency in MS individuals than in healthy controls in a study performed on 3 Italian families with MS members, 120 unrelated MS individuals, and healthy controls in order to identify new exonic low-allele frequency variants associated with MS [53]. Furthermore, in a GWAS on 776 MS individuals and 75 healthy controls conducted to study the gray matter pathology indicated by cortical thinning in MS, WWOX was noted to be associated with the thinning of different cortical regions in MS [54]. Additionally, in an analysis of genetic data from 15 different MS genome-associated studies of 47,429 MS individuals and 68,374 healthy controls, the rs12925972 WWOX variant was suggested to be significantly associated with MS susceptibility with a *p =* 3.07 × 10^−8^ and an OR of 1.099 [55]. Altogether, WWOX could play important roles in MS, perhaps due to its emerging function controlling myelination and oligodendrocyte differentiation.

### 6.3. WWOX in Autism Spectrum Disorders (ASD)

Heterozygous deletions and duplications overlapping the WWOX gene have been observed in individuals with ASD [56,57]. Copy number variants (CNVs) of WWOX have been reported in many ASD individuals characterized with less severe ASD phenotypes and intelligence quotient (IQ) levels approximate to the normal ranges, suggesting that heterozygous WWOX variants were a low-penetrance risk factor for ASD [58,59]. These WWOX variants act as weak risk factors and are generally associated with milder ASD phenotypes.

### 6.4. WWOX in Early Onset Epilepsy and Ataxia

Consistent with the observations in WWOX animal models, human patients with WWOX bi-allelic loss of function variants (deletions, nonsense, and some missense mutations) were found to be associated with autosomal recessive cerebellar ataxia, epilepsy, optic tract atrophy, retinal degeneration, growth retardation, developmental delay, intellectual disabilities, microcephaly with seizures, and early death [4,5,6,60,61,62,63,64,65,66,67,68,69,70,71,72,73,74,75,76]. The range of phenotypic severity of WWOX-deficient individuals extends from less severe phenotypes with later onset and non-progressive microcephaly, as seen in SCAR12, to more severe phenotypes with progressive microcephaly, seizures, global developmental delay, optic atrophy, and early age lethality, as observed in WOREE. Moreover, brain magnetic resonance images (MRI) of children with WOREE revealed an abnormally thin or hypoplastic corpus callosum, progressive optic atrophy, and brain atrophy as the most common signatures, while delayed myelination and white matter hyperintense signals were reported in some cases [5]. In patients with SCAR12, brain MRIs revealed mild hypoplasia of the cerebellum or cerebellar vermis. There was also a description of a patient with epilepsy of infancy with migrating focal seizures (EIMFS), in which the seizures started at 2.5 months with infantile spasms, and at the age of 6 years, there was profound severe intellectual impairment, microcephaly, spasticity, hypotonia, and scoliosis [73]. Another interesting West syndrome case described a heterozygous 6.8-Mb deletion of WWOX suggesting other genes within this large genomic region might be involved [77]. Although there are limited studies, overall, the phenotypic severity of WWOX deficient patients appears to be variant-specific, suggesting a phenotype–genotype correlation.

## 7. WWOX Variants and Mutations in Early Onset Epilepsy and Ataxia

Using Mutationmapper [78], we generated a gene map of all reported pathogenic WWOX variants demonstrating the range of variants spanning the gene (Figure 3). All SCAR12 individuals carry only missense variants (Figure 3 (red), Table 1), however, as only a small number of SCAR12 individuals have been reported to date, it remains unknown if this genotype is common to all SCAR12 patients or if it represents a sampling bias. This may be due to either the reserved protein function in the missense variants or the presence of patient-specific modifiers on the gene. Although the loss of function variants (nonsense, frameshift, splice-site) are more common, the missense variants are also observed in WOREE syndrome individuals. A better understanding of the genotype–phenotype correlations in the SCAR12 and WOREE syndromes will guide research efforts towards a better understanding of the underlying pathophysiology and treatment options of these disorders and assist in clinical management. We encourage clinicians and patients to register their variants and phenotypes on the WWOX foundation website (https://www.wwox.org, 1 April 2021) to facilitate further research on the WWOX gene.

To perform a comprehensive review of all reported WWOX variants, we combined in this review article publicly available data, our own database of patients (Appendix A), peer-reviewed publications in PubMed (https://pubmed.ncbi.nlm.nih.gov/, 1 April 2021), the online mendelian database (OMIM) (https://omim.org/, 1 April 2021), the DECIPHER database (https://decipher.sanger.ac.uk/, 1 April 2021), the ClinVar database (https://www.ncbi.nlm.nih.gov/clinvar/, 1 April 2021), the DGV database (http://dgv.tcag.ca/dgv/app/home, 1 April 2021), and finally data from gnomAD [79,80], to understand variant frequency in healthy controls.

### 7.1. Published WOREE Cases

In total, 45 variants were reported in 56 WOREE patient cases (Table 1). Of these, 34 were loss of function mutations, and 11 were missense variants. Of the reported WOREE cases described, 19 individuals had either a homozygous or compound heterozygous missense variant. Glutamine to proline substitution on amino acid 230 (Q230P) is an example of one of the missense variants, representing a very conserved amino acid residue change. This can explain the more severe disease phenotype even without a loss of function alleles, or a condition in which a missense mutation was actually causative of a loss of function effect.

### 7.2. Further Patient Details

Patient data regarding gender, ethnicity, seizure onset, and death are presented where available (Table 1; an extended Table 1 is provided as Appendix A). Variants were reported in individuals spanning over 20 countries and various ethnicities. To date, six individuals with SCAR12 variants have been reported in two families with seizures manifesting at less than 2 years of age [4]. WOREE syndrome individuals have many discreet mutations [5,12], with most mutations only described in a single patient (Table 1).

### 7.3. Data Collected from ClinVar

The dataset was divided into single-gene variations and continuous gene deletions, and subdivided into the five variation calls of benign, likely benign, variant of unknown significance (VUS), likely pathogenic, and pathogenic [81].

We found a total of 544 variants in WWOX deposited in ClinVar (https://www.ncbi.nlm.nih.gov/clinvar/, 1 April 2021), with 224 being benign or likely benign, and 320 being recorded as either variants of unknown significance (VUS), pathogenic, or likely pathogenic variants. These variants included the missense and nonsense point mutations, the partial and complete deletions, and the mutations affecting splicing. Of these variants, 205 were single-gene variants smaller than 500 kb deletions in size, with the remainder representing large deletions of the chromosome, removing multiple genes (including WWOX) at the locus. Because variants in ClinVar can be reported as individual allelic variants without requiring information of the other allele, variants whose second allele was not described were excluded from our study. Out of four variants associated with large deletions between 50 and 500kb, two were already published, and their descriptions appear in Appendix A. Two others were not well described, and represent single-allele deletions. A single variant with a deletion between 1 kb and 50 kb was classified as a VUS due to insufficient evidence regarding its pathogenicity, and, furthermore, that variant was excluded from the dataset because no data regarding zygosity was available for that patient. Five additional variants between 51 bp and 1 kb were omitted for the same reasons. Most of the variants reviewed were smaller than 51 bp.

We also compared 103 VUS (Appendix A) using the gnomAD database to further characterize these variants. Of these, 93 were missense variants, 6 intronic, 2 nonsense and 2 were indel. We found that 87% of the variants in gnomAD appeared between 1–109 times, with an average of 14 individual alleles per variant. None of these variants were seen in a homozygous state. In total there were 1057 alleles out of approximately 250 thousand alleles in gnomAD, giving an allele frequency of approximately 1:250. We do not know if all the variants are in cis or trans, i.e.: if they are on the same or different haplotypes. If they are indeed on different haplotypes, the expected frequency of disease cases was 1:62,000. If some are actually in cis, the expected disease frequency would be lower. We also assume random mating, but that might not always be the case. For a population of 7.5 billion people, this would result in approximately 121,000 cases of WOREE in the entire population. This number is higher than that of the cases reported in the population, which can be indicated by adding the 56 published patients to the additional 160 cases documented in ClinVar, (ClinVar has allele data and not zygosity data, so we assume all of these variants are combined into pairs). in conclusion, we are aware of a total of 217 reported worldwide cases of WWOX-related patients.

The difference between these two numbers obviously needs addressing. One possibility is that there is an under-diagnosis of WWOX-related neurological syndromes, which may be due to lack of reporting or partial/no genetic workup in cases of unexplained epilepsy. It also could be that many of these variants are benign, which should be addressed in future studies.

The likely-pathogenic and pathogenic alleles were then curated and compared to those of the previously-published individuals and their variants. The most represented type of variant was the loss of function variant, except for the two described families with SCAR12, which were missense variants [4]. Most of the individuals had compound heterozygous or homozygous variants, which is consistent in individuals with a clinical description matching WOREE or SCAR12. We conclude that pathogenic variants in WWOX are most likely recessive (bi-allelic) loss of function mutations, resulting in a non-functional or non-existent protein. This is in concordance with their gnomAD pLI score of 0 (probability of loss of function intolerance) [82].

### 7.4. Unpublished WWOX Cases

Using the DECIPHER and ClinVar databases, we identified ten previously unpublished individuals with pathogenic WWOX mutations (Table 2 and Appendix A). Four patients were found in the DECIPHER database and 6 in ClinVar. Two individuals identified in DECIPHER were described with epileptic encephalopathy, brain ventriculomegaly, cryptorchidism, hydrocele testis, and inguinal hernia, while the other two individuals were described with epileptic encephalopathy and profound global developmental delay.

Of the six cases identified in ClinVar, all individuals had the loss of function mutations and were described as having seizures. Two of the individuals had developmental delay, two had microcephaly, and one had dystonia and an adducted thumb.

It is worth mentioning, as noted previously by Piard et al. [5], that the 46,XY DSD phenotype described by White et al. [7] was not, due to a single heterozygous allelic deletion, noted in any of the published or unpublished cases. This patient harbored a heterozygous deletion of part of the WWOX gene and presented from birth with ambiguous genitalia, undifferentiated gonadal tissue, and immature testis.

## 8. Future Possibilities for Personalized Medicine Interventions

Antiepileptic drugs (AEDs) are used in the mainstream clinical management of WOREE individuals. However, they are not effective, with most individuals becoming refractory to treatment (Appendix A). The poor response of WOREE patients to AEDs challenges the scientific and clinical communities to develop alternative therapeutic approaches. Without exact genotype–phenotype correlations, clinical management is hampered by limited knowledge of the underlying pathophysiology of this disorder, the exact role of WWOX in the brain, and a lack of understanding of whether specific missense variants lead to the gain-of-function or the loss-of-function (LOF) of the WWOX protein.

The development of WWOX-brain organoid cultures and models could aid in this respect. Brain organoids are generated from patient cells (blood leukocytes or skin fibroblasts), through reprogramming into induced pluripotent stem cells (iPSCs). iPSCs have been used in the past to model pediatric neurodevelopmental disorders such as Rett syndrome [83]. The use of iPSC-derived brain organoids has been shown to provide important insights into neurodevelopment and disease mechanisms such as microcephaly and others [84,85], and is intended to serve in WWOX-related disorders as well [86]. This will enable a better understanding of the mechanism of action, and also provide an improved model for future interventions and the testing of candidate drugs and therapies.

Genetic approaches to correct or replace defective genes in monogenetic diseases are continuously developing, but are largely dependent on the nature of the gene and its associated variants. For example, variants occurring in either splice acceptor or splice donor sites that alter the natural splicing machinery of genes are considered severe mutations. Defective splicing due to a splice-site mutation can cause a complete loss of the function phenotype by removing part of the final protein product and/or affecting the overall coding sequence, which could cause a frame shift or premature termination. Both can also lead to the complete elimination of one copy of the gene through nonsense-mediated mRNA decay (NMD). Efforts to reduce or completely ablate the NMD process have recently shown promise [87]. This could be a possible therapy in patients with splice-site mutations such as the case of the Yemenite founder WWOX-splicing mutation [71], and also in patients with any NMD-based allele deletion.

The WWOX loss of function variants involving loss of WWOX protein expression suggest the possibility of introducing the coding region of WWOX by genetic editing or gene therapy. For individuals with complete loss of function mutations, a copy of the WWOX cDNA introduced into target cells could potentially provide a therapeutic option. Recent success in treating spinal muscular atrophy (SMA), a neuromuscular disorder with grave consequences of varying severity, has recently shown success in gene therapy clinical trials where patients with SMA were treated with a partial to complete restoration of the gene through the use of an AAV9 vector [88]. With advances in technology and a better understanding of the underlying pathophysiology of each WWOX variant, it is conceivable that gene therapy will also become available for WWOX-related disease patients with LOF mutations.

Another therapeutic approach is gene editing using the CRISPR-Cas9 system. This system is among other in situ correction gene editing options, holding promise even if there is a need for single-nucleotide correction. The most common mechanism of use for repair is the extraction of patient cells, ex vivo repair using CRISPR-Cas9, and the reimplantation of the corrected cells [89]. This model is not feasible for neurological disorders and hence alternatives to generate systems based on CRISPR-Cas9 that can cross the blood–brain barrier are being developed [90]. CRISPR-Cas9 gene-editing technology is being tested in clinical trials in vitro for Leber’s congenital amaurosis 10 (LCA10), which results in blindness, to correct the CEP290 gene, which is responsible for the disease [91,92,93,94]. Such development is an important step towards the use of gene therapy to tackle diseases with no available treatments. It is thus clear that several approaches for providing precision medicine to patients with WWOX-neurological disorders should be exploited and developed.

## 9. Concluding Remarks

Here we presented an updated review of the literature describing the WWOX gene in early-onset epilepsy and associated neurological disorders. There are two main medical diagnoses with direct relation to the human germline WWOX pathogenic mutations SCAR12 and WOREE. As the SCAR12 condition was reported only reported in two families (six patients), it qualifies as an ultra-rare condition. The pathogenic mutations described in this gene are missense mutations, which might explain the difference in phenotypes between SCAR12 and WOREE syndrome. This gives rise to the hypothesis that while the complete loss of function variants will cause WOREE syndrome, the missense variants can sometimes maintain some residual protein function, therefore giving rise to the SCAR12 phenotype. The single allele heterozygous loss of function mutations are not expected to lead to severe phenotypes, as was determined from the gnomAD score of pLI = 0. Since there are insufficient medical and genetic diagnostic facilities in undeveloped and developing countries, we assume that there will be more patients diagnosed with WWOX mutations worldwide. It is worthwhile to include the WWOX gene in routine genetic tests in order to provide treatment with precision medicine approaches, as discussed above. To summarize, the “red flags” for WWOX-related disorders should be suspected in children with early-onset refractory epilepsy, profound developmental delay, abnormal EEG patterns, and brain abnormalities, possibly with eye involvement. This combination, alongside an autosomal recessive inheritance pattern, should trigger this diagnosis in patients.

The fact that WWOX-related epilepsy disorders are rare requires a joint effort from the scientific community and patient advocates to achieve a better understanding of WWOX-related diseases and their underlying mechanisms to develop new curative therapeutics. Very recently, two platforms were generated to help reach all the patients and mutations identified globally. A dedicated WWOX Foundation (https://www.wwox.org, 1 April 2021) was established with the aim of increasing awareness, helping develop novel treatment options, and finding a cure for WWOX-related syndromes. A registry database will be established to follow all cases. Another platform exists on the Human Disease Genes website, which provides information on patients with germinal mutations in WWOX, including clinical data, molecular data, management, and research options (https://humandiseasegenes.nl/WWOX/, 1 April 2021).

Further knowledge of the protein structure, domains, and interacting proteins of WWOX will be required to enable a better understanding of the true nature of its mechanism of action. Using animal models to dissect WWOX function in the brain will be instrumental while developing brain organoid models from patient-derived iPSCs will enable a better understanding of the complex interactions between WWOX and other cellular factors in human brain tissue and will serve as an invaluable platform for testing new therapeutic options.

## Figures and Tables

**Figure 1 cells-10-00824-f001:**
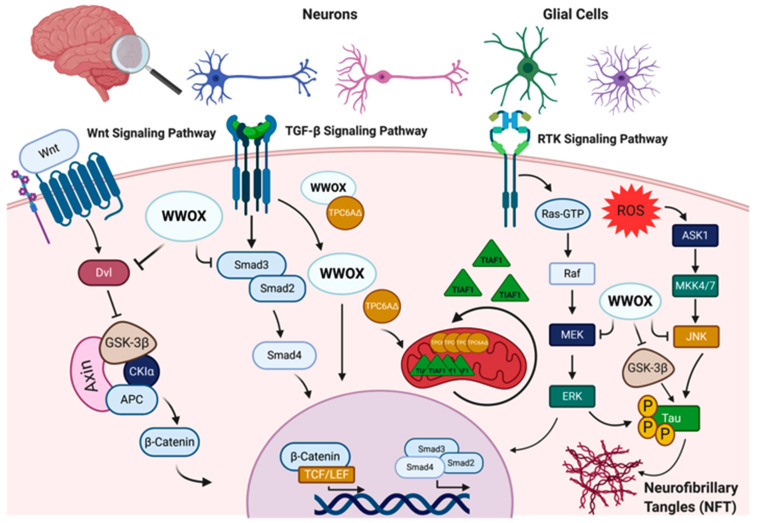
WW domain-containing oxidoreductase (WWOX) interaction partners and signaling pathways in the nervous system. WWOX protein–protein interactions and signaling in the nervous system involves: I. The Wnt/β-catenin signaling pathway, where WWOX negatively regulates this pathway through interaction with Dishevelled (Dvl) proteins and/or glycogen synthase kinase-3 β (GSK-3β). WWOX deficiency has been shown to be associated with increased nuclear β-catenin activity. II. WWOX has been also shown to modulate the transforming growth factor-β (TGF-β) signaling pathway through regulation of the SMAD proteins and the trafficking protein particle complex 6AΔ (TPC6AΔ). In the absence of WWOX, the SMAD/co-SMAD complex is inhibited, and TPC6AΔ and anti-apoptotic factor 1 (TIAF1) aggregate in the mitochondria, contributing to CNS pathologies. III. WWOX could also modulate the receptor tyrosine kinase (RTK) and c-Jun N-terminal kinase (JNK) signaling pathways to regulate tau phosphorylation and the formation of neurofibrillary tangles (NFT). Other abbreviations used: APC, adenomatosis polyposis; CKIα, casein kinase Iα; TCF/LEF, T-cell factor/lymphoid enhancing factor; ROS, reactive oxygen species; ASK1, apoptosis signal-regulating kinase 1; MKK4/7, mitogen-activated kinase kinase 4/7; RAF, rapidly accelerated fibrosarcoma; MEK: mitogen-activated protein Kinase; ERK, extracellular signal-regulated kinase. Figure was created via BioRender.

**Figure 2 cells-10-00824-f002:**
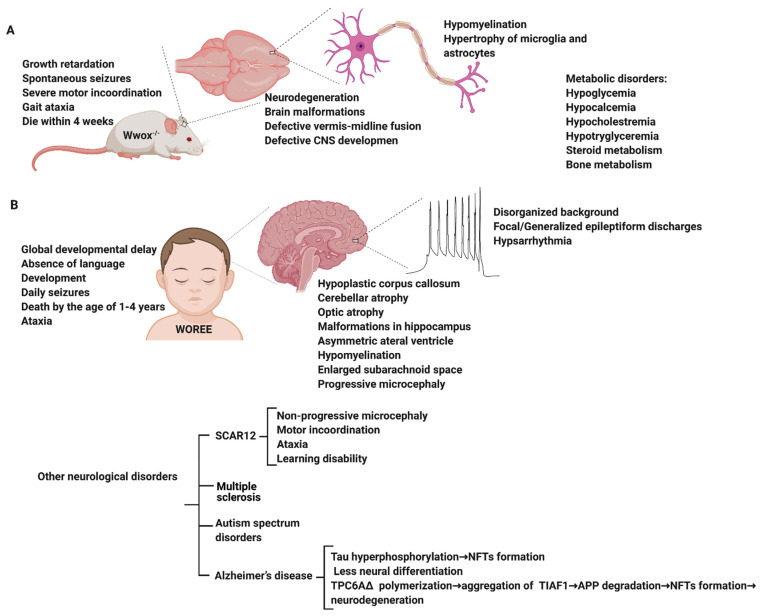
Summary of the neurological disorders associated with WWOX germline mutations. Upper panel (**A**) shows phenotypes associated with WWOX deficiency in rodents. Lower panel (**B**) summarizes neuropathological diseases associated with WWOX gene mutations in humans. NFTs, neurofibrillary tangles; TPC6AΔ, trafficking protein particle complex 6A; TIAF1, TGF-β1 induced antiapoptotic factor; APP, amyloid precursor protein. Figure was generated with BioRender.com.

**Figure 3 cells-10-00824-f003:**
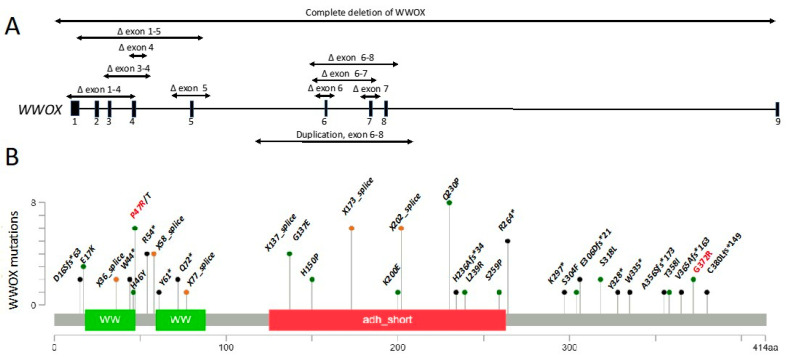
WWOX variants identified in patients with SCAR12 and WOREE syndrome. (**A**) Deletions leading to copy number variations in the WWOX gene. (**B**) Single-nucleotide variations (SNVs) leading to point mutations and protein changes. Mutation diagram circles are colored to indicate type of mutations: green for missense, orange for splice site, and black for truncating mutations including nonsense, frameshift deletion, and frameshift insertion. SCAR12 is mostly associated with missense mutations (red), while WOREE is associated with the deletions, missense, and nonsense mutations (black). Number of patients with increased frequency of a given mutation is indicated (height of the vertical lines).

**Table 1 cells-10-00824-t001:** Published WOREE and SCAR12 patients.

Name	Variant type	Protein change	Cases/references	Gender	Ethnicity	Zygosity	Seizure onset	Death
**Spinocerebellar ataxia, autosomal recessive 12**								
NM_016373.4(WWOX):c.139C>A (p.Pro47Thr)	Missense	P47T	4 siblings [4,8]	F/F/M/F	Saudi-Arabian	Homozygous	12m/9m/9m/9m	17-26 y
NM_016373.4(WWOX):c.1114G>C (p.Gly372Arg)	Missense	G372R	2 siblings [4]	M/F	Israeli	Homozygous	<2 y/<2y	5-10 y
**Developmental and epileptic encephalopathy 28**								
NM_016373.4(WWOX):c.160C>T (p.Arg54Ter)	Nonsense	R54*	1 [60]	F	Egyptian	Homozygous	2m	16m
WWOX-null alleles			2 [63]	F/M	Turkish		3 m/prenatal	22m/Mtp
GRCh37/hg19 16q23.1(chr16:78180603-78208482)x0	Deletion (Exon 5)		1 [61]	M	Emirati	Homozygous	2 w	5 m
NM_016373.4(WWOX):c.606-1G>A	Splicing variant		5 patients from 2 families [65]	F/F/F/M/F	Saudi Arabian	Homozygous	2m/3m/2m/2m/3m	<3 y/<3 y/<3 y/<3 y/<3 y
NM_016373.4(WWOX):c.46_49del (p.Asp16fs)	Deletion (Exon 1)/Nonsense	D16fs	2 [62]	M/F	Portuguese	Compound heterozygous	5 m/5m	alive at 4y/3 y
NM_016373.4(WWOX):c.140C>G (p.Pro47Arg)	Missense	P47R						
NM_016373.4(WWOX):c.1005G>A (p.Trp335Ter)	Nonsense	W335*	1 [62]	F	Portuguese	Compound heterozygous	7 w	24 m
NM_016373.2(WWOX):c.517-?_605+?del	Deletion							
NM_016373.3(WWOX):c.-366-?_516+?del	Deletion		1 [62]	F	Portuguese	Compound heterozygous	2m	alive at 3y
NM_016373.4:c.517_1056del	Deletion (Exon 6 to Exon 8)							
NM_016373.4(WWOX):c.889A>T (p.Lys297*)	Nonsense		1[62]	F	Portuguese	Compound heterozygous	2m	38m
deletion encompassing the WWOX locus (2.8 Mb del)	Deletion							
NM_016373.4(WWOX):c.131G>A (p.Trp44Ter)	Nonsense	W44*	2 [64]	F/F	Qatari	Homozygous	7 w/7w	7y/20 m
NM_016373.4(WWOX):c.918del (p.Glu306fs)	Deletion (Exon 8)/Nonsense	E306fs	1 [66]	M	Romanian	Compound heterozygous	4 w	<3 y
NM_016373.4(WWOX):c.173-1G>T	Splicing variant	Asp58Alafs*3						
NM_016373.4(WWOX):c.160C>T (p.Arg54Ter)	Nonsense	R54*	2 [93]	M/F	Saudi Arabian	Homozygous	2 m	alive at 3 m/18 m
NM_016373.4(WWOX):c.409+1G>T	Splicing variant		1 [93]	M	Saudi Arabian	Homozygous	2 m	alive at 21 m
GRCh37/hg19 16q23.1(chr16:78143675-78149052)	Deletion (Exon 3 to Exon 4)		1 [6]	M	Emirati	Homozygous	5 w	alive at 2 y
NM_016373.4(WWOX):c.606-1G>A	Splicing variant		1 [6]	M	Emirati	Homozygous	5 w	alive at 13 m
NM_016373.4(WWOX):c.689A>C (p.Gln230Pro)	Missense	Q230P	2 [68]	F/F	Afghan	Homozygous	2 m/2 m	alive at 12 y/10 y
NM_016373.4(WWOX):c.716T>G (p.Leu239Arg)	Missense	L239R	1 [94]	F	Turkish	Homozygous	2 m	NA
NM_016373.4:c.1_409del	Deletion (Exon 1 to Exon 4)		1 [5]	M	Pakistani	Homozygous	7 m	2 y
NM_016373.4:c.49G>A (p.Glu17Lys)	Missense	E17K	1 [5]	F	NA	Compound heterozygous	11 w	NA
NM_016373.4:c.911C>T (p.Ser304Phe)	Missense	S304F						
NM_016373.4(WWOX):c.160C>T (p.Arg54Ter)	Nonsense	R54*	1 [5]	F	French	Compound heterozygous	2.5 m	NA
NM_016373.4:c.1094_1095delTC (p.Val365AlafsTer163)	Deletion (Exon 9)/Nonsense	V365Afs*163						
NM_016373.4:c.231_409del (p.Asp77GlufsTer27)	Deletion (Exon 4)	D77Efs*27	1 [5]	F	Italian	Compound heterozygous	day 20	NA
NM_016373.4:c.607_791+1del	Deletion (Exon 7 to Intron 7)							
NM_016373.4(WWOX):c.410G>A (p.Gly137Glu)	Missense	G137E	1 [5]	M	NA	Compound heterozygous	day 1	NA
NM_016373.4(WWOX):c.953C>T (p.Ser318Leu)	Missense	S318L						
NM_016373.4(WWOX):c.410G>A (p.Gly137Glu)	Missense	G137E	1 [5]	M	British	Compound heterozygous	day 1	2 y6 m
NM_016373.4:c.517_791del	Deletion(Exon 6 to Exon 7 )							
NM_016373.4:c.517_791del	Deletion (Exon 6 to Exon 7)		2 [5]	M/M	French	Compound heterozygous	3 m/2.5m	NA/NA
NM_016373.4:c.449A>C (p.His150Pro)	Missense	H150P						
NM_016373.4:c.411_516+1del	Deletion(Exon 5 to Intron 5)		1 [5]	F	French	Compound heterozygous	day 1	6 m
NM_016373.4:c.607_791+1del	Deletion (Exon 7 to Intron 7)							
NM_016373.4(WWOX):c.689A>C (p.Gln230Pro)	Missense	Q230P	1 [5]	F	Moroccan	Homozygous	2 m	3 y 3 m
NM_016373.4(WWOX):c.689A>C (p.Gln230Pro)	Missense	Q230P	1 [5]	F	French	Compound heterozygous	3 w	NA
NM_016373.4:c.598A>G (p.Lys200Glu)	Missense	K200E						
NM_016373.4(WWOX):c.1073C>T (p.Thr358Ile)	Missense	T358I						
NM_016373.4(WWOX):c.689A>C (p.Gln230Pro)	Missense	Q230P	1 [5]	F	French	Compound heterozygous	3 m	NA
NM_016373.4:c.1138dupT (p.Cys380LeufsTer149)	Duplication/Nonsense	C380Lfs*149						
NM_016373.4(WWOX):c.689A>C (p.Gln230Pro)	Missense	Q230P	1 [5]	F	Iranian	Homozygous	2 m	NA
NM_016373.4(WWOX):c.790C>T (p.Arg264Ter)	Nonsense	R264*	2 [5]	F/M	Indian	Homozygous	7 w/NA	8y 11 m/5y 2m
NM_016373.4(WWOX):c.173-1G>T		Asp58Alafs*3	1 [5]	F	French	Compound heterozygous	day 20	NA
NM_016373.4(WWOX):c.517_1056dup	Duplication	His173_Met352dup						
NM_016373.4:c.517_1056del	Deletion (Exon 6 to Exon 8)		1 [5]	F	Caucasian	Compound heterozygous state	day 2	NA
NM_016373.4(WWOX):c.953C>T (p.Ser318Leu)	Missense	S318L						
NM_016373.4:c.517_1056del	Deletion (Exon 6 to Exon 8 )		1 [5]	M	SubSaharan	Homozygous	1 m	27 m
NM_016373.4:c.517_1056del	Deletion (Exon 6 to Exon 8)		1 [5]	F	British	Compound heterozygous	day 5	2y5m
NM_016373.4(WWOX):c.705dup (p.His236fs)	Frameshift							
NM_016373.2(WWOX):c.517-?_605+?del	Deletion		1 [70]	F	USA	Homozygous	2 w	6 y
NM_016373.4:c.517-2A>G	Splicing variant		4 [71]	F/F/F/M	Yemenite Jews	Homozygous	3m/3 w/1m/6w	30 m/33 m/9 m/alive at 6 m
NM_016373.4:c.517-2A>G	Splicing variant		2 [71]	M/F	Yemenite Jews	Compound heterozygous	2 w/3 w	alive at 4y/3 y
NM_016373.4(WWOX):c.689A>C (p.Gln230Pro)	Missense	Q230P						
NM_016373.4:c.173-2A>G	Splicing variant		1 [72]	M	Chinese	Compound heterozygous	day 19	NA
NM_016373.4:c.775T>C (p.Ser259Pro)	Missense	S259P						
NM_016373.3 (and GRCh37/hg19): deletion-chr16:78146639-78151289, chr16:78166192-78184119	Deletion		1 [73]	F	Australian	compound heterozygous	2.5m	alive at 7y
NM_016373.4:c.49G>A (p.Glu17Lys)	Missense	E17K						
GRCh37/hg19: 16q22.2q23.1(71,689,186–78,530,357)x1	6.8 Mb Deletion		1 [77]	M	Japanese	heterozygous	16m	4 y7m
NM_016373.4:c.229_230+2delGAGT	Deletion (Exon 3 to Intron 3 )		1 [75]	F	Chinese	Compound heterozygous	8 w	5 m
NM_016373.4:c.1065dupA (p.Ala356SerfsTer173)	Duplication/Nonsense	A356Sfs*173						
NM_016373.4(WWOX):c.790C>T (p.Arg264Ter)	Nonsense	R264*	2 [42]	M/M	Sicily Italy	Homozygous	45 days/prenatal	alive/Mtp
NM_016373.4(WWOX):c.984C>G (p.Tyr328Ter)	Nonsense	Y328*	1 [74]	M	Chinese	compound heterozygous	1 m	1y
NM_016373.4(WWOX):c.172+1G>C	Splicing variant							

**Table 2 cells-10-00824-t002:** Unpublished patients from DECIPHER and ClinVar databases.

Decipher ID	Location	Variant Change	Protein Change	Size	Cases	Zygosity	Pathogenicity	Inheritance	Sex
**407439**	78098478-78259430	Deletion		160.95 kb	2	Homozygous	Pathogenic	Biparental	46XY
**384984**	78133724-78133724	c.49G > A	E17K	SNV	2	Compound heterozygous	Likely pathogenic	Unknown	46XX
78022830-78216167	Deletion		193.34 kb	Likely pathogenic	Maternally inherited, constitutive in mother
**ClinVar**									
**Name**	**Variant change**	**Protein Change**	**Cases**	**Zygosity**	**Pathogenicity**	**Phenotype(s)**	**GRCh37Chromosome**	**GRCh37Location**	
**NM_016373.4(WWOX):c.108-2A>T**	Splice acceptor		1	Homozygous	Likely pathogenic	seizures, Developmental regression, global developmental delay, microcephaly, and Dystonia	16	78142318	
**NM_016373.4(WWOX):c.183C>A (p.Tyr61Ter)**	Nonsense	Y61*	1	Compound heterozygous	Likely pathogenic	a phenotype consistent with the WWOX-associated disease.	16	78143685	
**NM_016373.4(WWOX):c.918del (p.Glu306fs)**	Deletion (exon 8)	E306fs	Likely pathogenic	16	78466511	
**NM_016373.4(WWOX):c.214C>T (p.Gln72Ter)**	Nonsense	Q72*	1	Compound heterozygous	Pathogenicity	seizures	16	78143716	
**NC_000016.10:g.(?_78099759)_(78115174_?)del**	Deletion (exons 1–4)		Pathogenicity	16	78133656–78149071	
**NM_016373.4(WWOX):c.214C>T (p.Gln72Ter)**	Nonsense	Q72*	1	Compound heterozygous	Pathogenicity	microcephaly, infantile spasms/seizures	16	78143716	
**NC_000016.9:g.(?_78198060)_(78198206_?)del**	Deletion (exon 5)		Pathogenicity	16	78198060–78198206	
**NM_016373.4(WWOX):c.107+1G>A**	Splicing donor		1	Compound heterozygous	Pathogenicity	developmental delay and early-onset epileptic encephalopathy.	16	78133783	
**NM_016373.4(WWOX):c.136C>T (p.His46Tyr)**	Missinse	H46Y	Likely pathogenic	16	78142348	
**NM_016373.4(WWOX):c.409+1G>C**	Splice donor		1	Compound heterozygous	Likely pathogenic	seizures	16	78149052	
**NM_016373.4(WWOX):c.790C>T (p.Arg264Ter)**	Nonsense	R264*	Pathogenicity	16	78458951

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
