# Peer review of "Neurological Disorders Associated with WWOX Germline Mutations—A Comprehensive Overview"

_cells, 2021, doi:10.3390/cells10040824_

Round 1

Reviewer 1 Report

Reviewed manuscript titled “ Neurological disorders associated with WWOX germline mutations – A Comprehensive overview “ is a really comprehensive overview what is emphasised in the title, regarding non-cancer neurological disorders associated with WWOX germline mutations. It is really a big collection of WWOX individuals and generally in my opinion it is a high level of review article.

Every study in rare genetic disease is very important for good understanding of genotype and phenotype of patients affected by the orphan disorder with very rare occurrence.

I have only one minor comment described below in  Introduction:

In section “Clinical implications …” I propose to divided into separated subtitles:  Developmental and epileptic encephalopathy 28 and Spinocerebellar ataxia, autosomal recessive 12.

This manuscript should be published.

Author Response

Point-by-point response:

Reviewer 1:

Reviewed manuscript titled “ Neurological disorders associated with WWOX germline mutations – A Comprehensive overview “ is a really comprehensive overview what is emphasised in the title, regarding non-cancer neurological disorders associated with WWOX germline mutations. It is really a big collection of WWOX individuals and generally in my opinion it is a high level of review article.

Every study in rare genetic disease is very important for good understanding of genotype and phenotype of patients affected by the orphan disorder with very rare occurrence.

Response: We thank the reviewer for acknowledging the significance of our review article.

I have only one minor comment described below in  Introduction:

In section “Clinical implications …” I propose to divided into separated subtitles:  Developmental and epileptic encephalopathy 28 and Spinocerebellar ataxia, autosomal recessive 12.

Response: We thank the reviewer for this important point. We revised our manuscript accordingly.

This manuscript should be published.

Response

Reviewer 2 Report

The authors provide here a very comprehensive review on published and unpublished patients harboring allelic and, mostly, bi-allelic WWOX variants. 

They also discuss mechanisms of disease and other possible involvement of WWOX in other neurodevelopmental and neurodegenerative disorders. 

I appreciate this paper although I suggest some modifications.

Abstract: Please report here the number of WWOX patients previously reported in the literature according to the analysis made by authors as well as unpublished patients. Please also distinguish between DEE28/WOREE cases and SCAR14 cases in order to give an initial overview of the total number of known cases. 

Lines 51-52: the term EIEE has been replaced by DEE (developmental and epileptic encephalopathy). Please change it and also unify the nomenclature for DEE28 or WOREE in order to reduce the possibility to create confusion for the readers. 

Lines 52-54: the concept that delineates DEE is that in addition to the developmental encephalopathy, there is a crucial role for the abnormal EEG and resistant epilepsy in determining neurodegeneration. Thus, the definition of main features given at lines 52-54 is not completely correct and should be revised.

Lines 52-54: how many DEE-causing genes have currently been identified? And how many are currently listed in OMIM?

Line 56: specify here DEE28 and WOREE.

Line 56: specify here, as done in the abstract, that bi-allelic germline variants…

Different lines: please use Italics when referring to WWOX gene

Line 67: this sentence is redundant. Please delete it or re-write it in order to make it different from what reported few lines above.

Line 66: usefulness of this paragraph is not clear. It seems to be an extension of the introduction. If the aim of the authors is to group here the main clinical features of WWOX-related disorders, I suggest to be more precise: since West syndrome is yet a type of epileptic encephalopathy, it is worthwhile to add some details that may be useful to distinguish the two types of DEE (WWOX-related West syndrome versus DEE28), specifically with regard to seizures’ onset and semiology and EEG. Please also describe these two forms first and then SCAR14.

Also, the usefulness to review published and unpublished cases with a specific disorder is, among others, to underlie possible clues for diagnosis: are there some clues (clinical, neuroradiological, EEG, exc..) that can be used to identify or to suspect WWOX-related disorders?

The authors should discuss somewhere in the paper if DEE28 has some specific features respect to the remaining DEEs or if SCAR14 has some specific features respect to other SCARs.

How many autosomal recessive and autosomal dominant early-onset ataxias are associated with epilepsy?

All these data are really informative ; otherwise the phenotype of WWOX is quite well known.

Line 80: delayed myelination appears as white matter hyperintensity (in T2 images), thus the authors should be more precise in describing the brain MRI of these patients.

Paragraph WWOX in childhood epilepsy – lines 206-219: This paragraph seems that it takes exactly the information already mentioned several times before. It is redundant and it contributes to overstretch the article.

The different forms of epilepsy/DEE have been cited several times but it seems that the authors have never reported the Epilepsy of Infancy with Migrating Focal Seizures reported in Ann Neurol. 2019 Dec;86(6):821-831. doi: 10.1002/ana.25619.

Additionally: why the authors use (several times) the term “childhood” since in most cases the onset of epilepsy in WWOX related disorders is set in infancy. I would not use any “timing” here and elsewhere (es line 206, line 220),  but only something as “WWOX in epileptic disorders”.

Paragraph lines 220-245: the authors also include patients with SCAR14, thus the name of the paragraph itself should be revised.

Figure 3: Are there the (previously cited) patients with West syndrome and also the cases with migrating focal epilepsy of infancy included in the figure 3?

Line 248: references are needed here

Line 318: please change ASM with AEDs (antiepileptic drugs)

Lines 318-320: the section on symptomatic treatment is really poor, with only three AEDs cited. Please add much more info on (symptomatic) treatments of these patients. Also, at line 319 the authors say that “..some individuals becoming refractory..”. However, DEEs (as well as West syndrome or epilepsy with migrating focal seizures of infancy) are all drug resistant seizures disorders. So please specify how many WWOX patients have drug resistant epilepsy? Does drug resistance reduce with time?

Table 1 and references: I have noticed that few papers have not been included in the authors’ analysis. I think that an effort to include all available papers must be made in order to make this article the most updated review on WWOX-related disorders.

The references I found to be not included are the following:

Burgess et al. Ann Neurol. 2019 Dec;86(6):821-831. doi: 10.1002/ana.25619.

Iacomino et al. Front Neurosci. 2020 Jun 11;14:644. doi: 10.3389/fnins.2020.00644. eCollection 2020.PMID: 32581702 

Mori et al Brain Dev. 2019 Nov;41(10):888-893. doi: 10.1016/j.braindev.2019.07.005. Epub 2019 Jul 25.PMID: 31353122

Ehaideb et al. Transl Neurosci. 2018 Dec 31;9:203-208. doi: 10.1515/tnsci-2018-0029. eCollection 2018.PMID: 30746283

Author Response

Reviewer 2:

Comments and Suggestions for Authors

The authors provide here a very comprehensive review on published and unpublished patients harboring allelic and, mostly, bi-allelic WWOX variants. 

They also discuss mechanisms of disease and other possible involvement of WWOX in other neurodevelopmental and neurodegenerative disorders. 

I appreciate this paper although I suggest some modifications.

Response: We thank the reviewer for acknowledging the significance of our review article.

Abstract: Please report here the number of WWOX patients previously reported in the literature according to the analysis made by authors as well as unpublished patients. Please also distinguish between DEE28/WOREE cases and SCAR14 cases in order to give an initial overview of the total number of known cases. 

Response: Numbers of patients were added as requested.

Lines 51-52: the term EIEE has been replaced by DEE (developmental and epileptic encephalopathy). Please change it and also unify the nomenclature for DEE28 or WOREE in order to reduce the possibility to create confusion for the readers. 

Response: We thank the reviewer for this important point. The term EIEE has been replaced throughout the entire manuscript.

Lines 52-54: the concept that delineates DEE is that in addition to the developmental encephalopathy, there is a crucial role for the abnormal EEG and resistant epilepsy in determining neurodegeneration. Thus, the definition of main features given at lines 52-54 is not completely correct and should be revised.

Response: We thank the reviewer for this important point. The definition of DEE has been corrected and a citation was added.

Lines 52-54: how many DEE-causing genes have currently been identified? And how many are currently listed in OMIM?

Response: We thank the reviewer for this suggestion. This information was added to the manuscript

Line 56: specify here DEE28 and WOREE.

Response: We specify per request of the reviewer.

Line 56: specify here, as done in the abstract, that bi-allelic germline variants…

Response: We specify per request of the reviewer.

Different lines: please use Italics when referring to WWOX gene

Response: We thank the reviewer for this notion. We now corrected as ROMA/ITALIC when referring to human WWOX gene, ROMAN when referring to WWOX protein. When referring to mouse gene, we use the standard of Wwox in italics.

Line 67: this sentence is redundant. Please delete it or re-write it in order to make it different from what reported few lines above.

Response: We agree that this is redundant and hence this sentence was deleted.

Line 66: usefulness of this paragraph is not clear. It seems to be an extension of the introduction. If the aim of the authors is to group here the main clinical features of WWOX-related disorders, I suggest to be more precise: since West syndrome is yet a type of epileptic encephalopathy, it is worthwhile to add some details that may be useful to distinguish the two types of DEE (WWOX-related West syndrome versus DEE28), specifically with regard to seizures’ onset and semiology and EEG. Please also describe these two forms first and then SCAR14.

Response: We thank the reviewer for this suggestion. We incorporated the changes suggested by the reviewer and enhanced the description of the different conditions.

Also, the usefulness to review published and unpublished cases with a specific disorder is, among others, to underlie possible clues for diagnosis: are there some clues (clinical, neuroradiological, EEG, exc..) that can be used to identify or to suspect WWOX-related disorders?

Response: We thank the reviewer for this suggestion. This was added to the conclusion.

The authors should discuss somewhere in the paper if DEE28 has some specific features respect to the remaining DEEs or if SCAR14 has some specific features respect to other SCARs.

Response: We thank the reviewer for this suggestion. This was added as advised.

How many autosomal recessive and autosomal dominant early-onset ataxias are associated with epilepsy?

Response: We thank the reviewer for this suggestion. This was added in the text

All these data are really informative ; otherwise the phenotype of WWOX is quite well known.

Line 80: delayed myelination appears as white matter hyperintensity (in T2 images), thus the authors should be more precise in describing the brain MRI of these patients.

Response: Per request of the reviewer this was clarified in the text.

Paragraph WWOX in childhood epilepsy – lines 206-219: This paragraph seems that it takes exactly the information already mentioned several times before. It is redundant and it contributes to overstretch the article.

Response: We thank the reviewer for this suggestion. We added content to the paragraph per request of the reviewer.

The different forms of epilepsy/DEE have been cited several times but it seems that the authors have never reported the Epilepsy of Infancy with Migrating Focal Seizures reported in Ann Neurol. 2019 Dec;86(6):821-831. doi: 10.1002/ana.25619.

Response: We apologize for missing this citation. This citation including a description of the WWOX-related case was added.

Additionally: why the authors use (several times) the term “childhood” since in most cases the onset of epilepsy in WWOX related disorders is set in infancy. I would not use any “timing” here and elsewhere (es line 206, line 220),  but only something as “WWOX in epileptic disorders”.

Response: Per request of the reviewer childhood was changed to early onset

Paragraph lines 220-245: the authors also include patients with SCAR14, thus the name of the paragraph itself should be revised.

Response: Per request of the reviewer Ataxia was added to the name of the paragraph

Figure 3: Are there the (previously cited) patients with West syndrome and also the cases with migrating focal epilepsy of infancy included in the figure 3?

Response: In Figure3 we have all the variants that we are showing in table 1 and table 2 (published and unpublished patients)

Line 248: references are needed here

Response: Per request of the reviewer we added the references within the table

Line 318: please change ASM with AEDs (antiepileptic drugs)

Response: Per request of the reviewer we changed the name as guided.

Lines 318-320: the section on symptomatic treatment is really poor, with only three AEDs cited. Please add much more info on (symptomatic) treatments of these patients.

Response: We thank the reviewer for this suggestion. As the detailed information regarding most of the published cases of WWOX-related disorders is lacking, without any mention of the treatment they received, and it seems the discussion regarding which epilepsy treatment they received is futile, as no treatment actually works, we omitted the specific AEDs and continued the discussion without this. A complete discussion regarding treatment of epilepsy with AEDs and which AED is most suitable is out of scope for this manuscript.

Also, at line 319 the authors say that “..some individuals becoming refractory..”. However, DEEs (as well as West syndrome or epilepsy with migrating focal seizures of infancy) are all drug resistant seizures disorders. So please specify how many WWOX patients have drug resistant epilepsy? Does drug resistance reduce with time?

Response: Based on published data there are 23 WOREE patients who displayed drug resistant epilepsy (indicated in Table 1).

Table 1 and references: I have noticed that few papers have not been included in the authors’ analysis. I think that an effort to include all available papers must be made in order to make this article the most updated review on WWOX-related disorders.

The references I found to be not included are the following:

Response: Per request of the reviewer, we added all references.